# Targeting Inflammasome Activation in Viral Infection: A Therapeutic Solution?

**DOI:** 10.3390/v15071451

**Published:** 2023-06-27

**Authors:** Chuan-Han Deng, Tian-Qi Li, Wei Zhang, Qi Zhao, Ying Wang

**Affiliations:** 1State Key Laboratory of Quality Research in Chinese Medicine, Institute of Chinese Medical Sciences, University of Macau, Avenida da Universidade, Taipa, Macao 999078, China; 2State Key Laboratory of Quality Research in Chinese Medicines, Macau University of Science and Technology, Av. Wai Long, Taipa, Macao 999078, China; 3Cancer Centre, Institute of Translational Medicine, Faculty of Health Sciences, University of Macau, Taipa, Macau 999078, China; 4MoE Frontiers Science Center for Precision Oncology, University of Macau, Avenida da Universidade, Taipa, Macao 999078, China; 5Department of Pharmaceutical Sciences, Faculty of Health Science, University of Macau, Avenida da Universidade, Taipa, Macao 999078, China; 6Minister of Education Key Laboratory of Tumor Molecular Biology, Jinan University, Guangzhou 510632, China

**Keywords:** inflammasome, viral infection, inflammatory response, innate immunity, host defense, therapy

## Abstract

Inflammasome activation is exclusively involved in sensing activation of innate immunity and inflammatory response during viral infection. Accumulating evidence suggests that the manipulation of inflammasome assembly or its interaction with viral proteins are critical factors in viral pathogenesis. Results from pilot clinical trials show encouraging results of NLRP3 inflammasome suppression in reducing mortality and morbidity in SARS-CoV-2-infected patients. In this article, we summarize the up-to-date understanding of inflammasomes, including NLRP3, AIM2, NLRP1, NLRP6, and NLRC4 in various viral infections, with particular focus on RNA viruses such as SARS-CoV-2, HIV, IAV, and Zika virus and DNA viruses such as herpes simplex virus 1. We also discuss the current achievement of the mechanisms involved in viral infection-induced inflammatory response, host defense, and possible therapeutic solutions.

## 1. Introduction

The innate immune system acts as the first barrier to defend against viral infection, initiate host defenses to promote viral clearance, or mediate the switch to an inflammatory response to trigger pathogenesis. Inflammasome assembly is one of the most important steps in innate immunity and over activation of innate immunity is a direct cause related to persistent morbidity and mortality in pathogenic viral infections. Due to the enormous number of new infections and deaths caused by notorious viruses such as severe acute respiratory syndrome coronavirus 2 (SARS-CoV-2), determining the central function of inflammasome activation in viral infections has attracted attention from all disciplines around the world. In this article, we discuss the involvement of several similar but distinct inflammasomes, including NOD-like receptor (NLR) family pyrin domain containing 3 (NLRP3), absent in melanoma 2 (AIM2), NLRP1, NLRP6, NLRP9b, NLR family CARD domain containing 4 (NLRC4), and caspase recruitment domain family member 8 (CARD8) in the infection of different types of viruses, such as human immunodeficiency virus (HIV), influenza virus 1 (IAV), SARS-CoV-2, and others. Each of these proteins has conserved domains to form the inflammasome and unique sequences to ensure specificity (Figure 1, Table 1). A better understanding of how viruses activate inflammasomes will facilitate the discovery and invention of targeted therapies that could offer alternative treatment for serious diseases triggered by viral infection (Table 2).

## 2. NLRP3 Inflammasome and SARS-CoV-2

### 2.1. The NLRP3 Inflammasome

The NLRP3 inflammasome is a well-studied inflammasome activated by diverse signals [1]. It is rapidly and exclusively assembled by the cytosolic proteins NIMA related kinase 7 (NEK7), NLRP3, PYD, and CARD domain containing protein (ASC) and pro-caspase-1 upon stimulation [2]. The activated NLRP3 inflammasome then leads to the cleavage of gasdermin D (GSDMD) and pro-caspase-1. The N-terminal of cleaved GSDMD translocates and inserts into the cell membrane and forms large pores of approximately 21 nm in diameter to disrupt the cell membrane and induce pyroptotic death. Mature caspase-1 further cleaves proinflammatory cytokines interleukin (IL)-18 and IL-1β, which are then released through the pore formed by GSDMD. Other inflammasomes, with the exception of NLRC4, assemble in a similar manner (Figure 2).

The primary function of IL-1β and IL-18 is to promote the stimulation of T cells and macrophages in the innate immune response. However, elevated serum levels of IL-1β and IL-18 also correlate with the severity of disease in SARS-CoV-2-infected patients [3]. Activated inflammasomes, including the NLRP3 inflammasome, are most commonly observed in immune cells involved in innate immunity such as macrophages. Inflammasomes such as the NLRP10 inflammasome also assemble in response to mitochondrial damage in other cell types, for example keratinocytes [4], but this is beyond the scope of this article.

### 2.2. The NLRP3 Inflammasome Assembly and Activation in SARS-CoV-2-Triggered Innate Immunity

SARS-CoV-2, a positive-sense single-stranded RNA virus, belongs to the *Betacoronavirus* genus of the Coronaviridae family. Its genome encodes viral spike (S), membrane (M), envelope (E), and nucleocapside (N) proteins [5]. The SARS-CoV-2 genome also encodes other viral proteins necessary to support its lifecycle. During the initial activation of innate immunity, SARS-CoV-2 virions are recognized by pathogen-associated molecular patterns (PAMPs) or damage-associated molecular patterns (DAMPs). Once PAMPs and DAMPs are activated and signaling has been passed through cell surface receptors, mainly including nucleotide-binding oligomerization domain NLRs, C-type lectin receptors, toll-like receptors (TLRs), and immune cells, especially monocytes, will produce proinflammatory cytokines and chemokines to fight back the infection of SARS-CoV-2 and clear infected cells [6].

Several SARS-CoV-2 viral proteins trigger priming of the NLRP3 inflammasome facilitated by different host factors. The S protein activates the canonical pathway controlling the NLRP3 inflammasome in both mononuclear cells and neutrophils in patients [7]. The non-structural protein (NSP)2 triggers the transcription of NLRP3 protein through NF-κB signaling [8]. The N protein of SARS-CoV-2 directly binds to NLRP3 and triggers the formation of the NLRP3 inflammasome, thereby inducing excessive inflammatory response, acute lung injury, and accelerated sepsis-induced death [9]. The SARS-CoV-2 NSP6 directly interacts with the vacuolar ATPase proton pump component ATP6AP1, suppressing the subsequent activation driven by its cleavage [10]. The interaction of NSP6 with ATP6AP1 and the downstream activation of the NLRP3 inflammasome is abrogated in SARS-CoV-2, expressing the L37F NSP6 variant [10]. SARS-CoV-2 stimulates activation of NLRP3 and associated inflammation through interacting with the Fcγ receptor on monocytes and macrophages instead of through binding to ACE2 on endothelial and epithelial cells [11]. The NLRP3 inflammasome in myeloid cells is activated by double-stranded DNA (dsDNA) released from infected epithelial cells. The subsequently released IL-1β from activated myeloid cells triggers IL-6 production in both immune cells such as myeloid and barrier tissue such as epithelial cells, which amplifies the proinflammatory circuit with subsequent activation of NLRP1, NLRP3, and AIM2 inflammasomes [12]. Circulatory exosomes obtained from patients in severe COVID-19 serve as an amplification hub for NLRP3 priming signals under this condition [13] (Figure 3). These findings suggest that SARS-CoV-2 infection triggers activation of multiple inflammasomes independently of priming signals such as TLR4.

### 2.3. Sustained Activation of the NLRP3 Inflammasome in SARS-CoV-2 Infection

Despite the host innate immune response to clear virus-infected cells, SARS-CoV-2 evolves evasion mechanisms to escape from interferon (IFN) signaling-mediated immune surveillance [14]. For example, the N protein binds to the linker region of GSDMD and blocks its cleavage in patients infected with SARS-CoV-2 [15]. The S protein promotes the secretion of IL-1β in macrophages obtained from SARS-CoV-2 infected patients. The S protein itself is able to elevate the NLRP3 cascade in the absence of other priming signals [16]. The chronic inflammation of the patients contributes to the persistent activation of the NLRP3 inflammasome in SARS-CoV-2 infection, which may explain the relatively higher disease severity rate in patients with metabolic diseases such as type 2 diabetes [17,18], chronic liver disease [19], and acute myocardial infarction [20]. In contrast, patients taking immunosuppressive drugs also have weak immune surveillance activity. For example, patients with chronic inflammatory bowel disease taking anti-TNF drugs such as infliximab are also more susceptible to SARS-CoV-2 infection even after three doses of vaccine [21]. This clinical evidence suggests that the evasion mechanisms of SARS-CoV-2 must be complicated and much remains to be explored.

Once SARS-CoV-2 successfully antagonizes immune surveillance and establishes prolonged infection, the NLRP3 inflammasome and signaling cascade is constitutively switched on and triggers cytokine storm, the massive production of proinflammatory cytokines [22]. Activation of the NLRP3 inflammasome in lung-resident monocytes such as neutrophils and macrophages in SARS-CoV-2-infected patients is the leading cause of systemic inflammation, subsequent acute lung injury, morbidity, or even mortality [11,23].

### 2.4. Regulation of the NLRP3 Inflammasome

There are normally two steps in the activation of NLRP3, namely the priming step or signal 1 and the activation phase or signal 2. Transcription and post-translational modifications of NLRP3 usually occur in the priming step [24]. Knockdown of TLRs in experimental mice inhibits NF-κB activity and downregulates the expression level of NLRP3.

The post-translational modifications generally occur during the stimulation with signal 1 to activate NLRP3, including phosphorylation, sumoylation, ubiquitination, etc. For example, the Ser198 site of NLRP3 is typically phosphorylated by JNK1, which induces NLRP3 inflammasome assembly and initiates the priming step [25]. Mitochondrial-anchored protein ligase acts as a SUMO E3 ligase, bridging the interaction between NLRP3 to SUMO-2/-3 ligase [26]. UBC9 is a SUMO E2 ligase that mediates sumoylation of NLRP3 on K689. Sumoylated NLRP3 lacks the binding site to deconjugating enzyme, so activation efficiency is reduced [27]. Different E3 ubiquitin ligases, including tripartite motif containing 31 (TRIM31) and Pellion2, have been identified and exhibit distinct effects on NLRP3 activation [28,29]. TRIM31 binds to NLRP3 and triggers its subsequent proteasome-mediated degradation through K48-linked polyubiquitination. An increased level of TRIM31 decreases IL-1β secretion, thereby reducing the inflammatory response [28]. Pellion2, on the other hand, promotes K63-linked polyubiquitination of NLRP3 upon stimulation with signal 1 [29]. The transcription of NLRP3 itself and pro-IL-1β that is required for the full activation of NLRP3 inflammasome all depends on the TLR [24]. Unfortunately, most of the factors involved in the post-translational modification of NLRP3 are lack of specificity and are not druggable. Several studies use RNA sequencing method to understand the complicated regulation of NLRP3 inflammasome [30]. However, the assembly and activation of inflammasome mostly require changes in protein-protein interaction rather than transcriptional regulation, and the results of the RNA sequencing study may not reflect the initial changes in post-translational modification or protein-protein interaction. 

The ORF3a protein encoded by SARS-CoV-2 is now known to fully activate both of these signals [31]. ORF3a is a highly conserved viral pore protein in SARS-CoV-2 that promotes self-cleavage and activation of pro-caspase 1. Meanwhile, ORF3a triggers the expression mechanism of NF-κB, leading to a decreased abundance of IκBα after massive phosphorylation and promotes binding of NF-κB to the promoter region of IL-1β, resulting in increased IL-1β expression [32]. The NF-κB p50 subunit accumulates in the nucleus of cells, expressing ORF3a [33]. NF-κB1 p50 is produced by the proteolysis of NF-κB1 p105 by the E3 ubiquitin ligase KPC1 [34]. ORF3a promotes E3 ligase TNF receptor-associated factor 3 (TRAF3)-mediated ubiquitination and activates p105 and ASC, which contain a caspase recruitment domain that activates NLRP3 [33,35]. In the presence of TRAF3, ORF3a forms a complex with TRAF3 and ASC to promote the TRAF3-mediated K63-linked ubiquitination of ASC, which is essential for activation of the NLRP3 inflammasome [35]. Therefore, antiviral treatment to reduce viral protein would largely eliminate direct activation of the NLRP3 inflammasome.

### 2.5. Targeting the NLRP3 Inflammasome to Control SARS-CoV-2 Pathogenesis

Because the NLRP3 inflammasome is activated exclusively in infected cells, it has become a hot target for the development of novel therapies. Results from a pilot study demonstrate that blocking NLRP3 inflammasome signaling exhibits potency in reducing COVID-19 immunopathology (Table 1). The combined treatment of oseltamivir and sirolimus represses the NLRP3 inflammasome activation and reduces inflammation in lung tissue in influenza virus H1N1 infection-induced lung injury [36]. The well-studied NLRP3 inhibitor, MCC950, ameliorates systemic inflammation in hACE2 transgenic mice infected with SARS-CoV-2 [37]. Metformin treatment suppresses TLR-induced mtDNA synthesis by targeting respiratory complex I, thus inhibiting activation of the NLRP3 inflammasome in SARS-CoV-2-infected mice and patients [38]. The monoclonal antibody of IL-1β, canakinumab, could suppress NLRP3 inflammasome signaling and significantly reduce systemic inflammation in type 2 diabetes patients infected with SARS-CoV-2 [17]. The FDA-approved medication disulfiram attenuates pore formation by the N-terminal GSDMD and prevents the release of proinflammatory cytokines upon activation of the NLRP3 inflammasome [39]. Treatment with disulfiram blocks SARS-CoV-2-triggered release of neutrophil extracellular traps (NETs) and reduces NETs-associated inflammation and subsequent organ damage [40]. Other NLRP3 specific inhibitors, such as OLT1177 and DFV890, are under clinical evaluation to reduce COVID-19 symptoms, early cytokine release syndrome, and respiratory function [2,41].

Nevertheless, some of the NLRP3 inhibitors exhibit efficacy in the treatment of SARS-CoV-2-infected patients. The timing of treatment with inflammasome inhibitors is critically important to overall efficacy because of the dual function of the NLRP3 inflammasome in innate immunity and persistent inflammation. Early intervention can shut down effective innate immunity. Therefore, rapid detection methods are required to determine the “sweet spot” for initiating treatment to prevent hyper-inflammation. Serum IL-1β level is not a qualified marker due to its extremely short half-life [42], which could only be detected in severe COVID-19 patients but often not even in IL-1-driven diseases [43]. The hallmark of inflammasome activation, the formation of microscopically visible ASC specks, could only be examined by confocal microscopy or imaging flow cytometry, which require robust instrument setup and are time-consuming [11]. Activation of multiple inflammasomes has been demonstrated in persistent infection [44], so it would not be surprising that inhibition of a single inflammasome or cytokine is unsatisfactory [17]. Co-infection and metabolites also hinder the efficacy of NLRP3 inhibitors. Epithelial infection with rhinovirus, a non-enveloped positive-strand RNA virus of the picornavirus family, leads to impaired IFN response and promotes prolonged SARS-CoV-2 infection through activation of retinoic acid-inducible gene I (RIG-I) inflammasome in chronic asthmatic patients [45]. Even a ketogenic diet counteracts the exacerbation of SARS-CoV-2 infection by suppressing NLRP3 signaling in aged mice [46].

## 3. NLRP3 Inflammasome in RNA Viral Infection

### 3.1. HIV

HIV is a type of lentivirus that belongs to the family of Retroviridae. Early detection of HIV infection is extremely challenging and a major barrier to eliminating the virus [47]. Similar to the NLRP3 inflammasome, assembly of the CARD8 inflammasome through autoproteolytic activation triggers pyroptotic cell death in human T cells [48]. The HIV protease protein cleaves CARD8 in the N-terminus to trigger the assembly of CARD8 inflammasome in a proteasome-dependent manner, which subsequently results in rapid pyroptotic cell death in HIV-infected macrophages and CD4^+^ T cells [49]. Thus, assembly of the CARD8 inflammasome is considered to be used as a sensor for HIV infection, as CARD8 inflammasome is also present in human primary acute myeloid leukemia samples or expression of 3CL protease of coronavirus and picornavirus [50,51]. Thus, CARD8 can only serve as a broad sensor of viral protease activity to determine HIV infection in combination with other markers. Pharmacological inhibition of HIV protease activity by bortezomib or dipeptidyl-peptidase 9 (DDP9) restricts activation of the CARD8 inflammasome and decreases killing of primary CD4^+^ T cells [49,52]. In contrast, activation of CARD8 with the DDP9 inhibitor Val-borPro synergizes with non-nucleoside reverse transcriptase inhibitors to illuminate HIV-infected CD4^+^ T cells in vitro and in humanized mice [53]. The controversial role and manipulation of CARD8 in HIV or other viral infections needs to be further explored in clinical trials.

Chronic HIV infection reduces CD4+ T cells counts and is the direct cause of HIV pathogenesis [54]. Only 5% of those CD4+ T cells undergo apoptosis, while the remaining 95% depend on assembly of the NLRP3 inflammasome and the following caspase 1-driven pyroptotic cell death. NLRP3 inflammasome activated in HIV-infected cells and associated production of ROS induces pyroptosis in CD4+ T cells, which is also one of the major reasons for the decreased efficacy of anti-HIV regimens in chronic infection [54]. Gene-set enrichment analysis of RNA from peripheral CD4+ T cells of antiretroviral therapy-treated HIV patients reveals that activation of the NLRP3 inflammasome is closely related to the maintenance of the HIV infection reservoir during treatment [55]. Treatment with the NLRP3 inhibitor MCC950 markedly reduces death of HIV-infected peripheral blood CD4+ T cells [54]. Caspase-1 inhibitor VX-765 also markedly decreases viral load, systemic inflammation, and death of CD4+ T cells in humanized NSG mice infected with HIV [56]. No combination of antiviral therapy with NLRP3 inhibitors has been reported in the HIV infection.

### 3.2. IAV

IAV has a single-stranded RNA genome and is the major cause of seasonal epidemics and sporadic pandemics. IAV infection is self-limiting for most healthy individuals, but it still leads to approximately 250,000 hospitalizations and 19,000 deaths in the USA alone during the 2022–2023 season (https://www.cdc.gov/flu/weekly/index.htm, accessed on 23 June 2023). Diverse mechanisms have been reported on how IAV infection activates host inflammatory response. The myxovirus resistance gene 1 product protein MxA is recognized as a restriction factor for IAV infection [57]. A recent study identified that MxA senses the IAV nucleoprotein, which binds to ASC and induces oligomerization of ASC and subsequent assembly of the inflammasome in respiratory epithelium [58]. M2 proton channel activity is pivotal for trans-Golgi network dispersion activated NLRP3 inflammasome upon IAV infection [59]. The DEAD-box helicase 3 X-linked coordinates assembly of the NLRP3 inflammasome when IAV NS1 protein expression is detected in myeloid cells, which is critical for IAV infection-mediated immune evasion [60].

Activation of the NLRP3 inflammasome drives MLKL-dependent necroptosis in IAV-infected cells. When lacking the expression of MLKL and NLRP3 inflammasome, infected cells turn to caspase-8-dependent apoptotic death [61]. Caspase-6 triggers assembly of the NLRP3 inflammasome; however, knockout of caspase-6 is the dispensable affect priming of NLRP3 or viral replication in IAV-infected BMDMs. Deficiency of caspas6 6 blocks IAV-induced different forms of cell death in BMDMs, prolongs survival, and reduces lung injury in IAV-infected mice. Both the N- and C-terminal domains of caspase 6 contribute to the formation of the caspase 6-Z-DNA binding protein 1 (ZBP1)-receptor interacting serine/threonine kinase 3 (RIPK3)-RIPK1 complex, which is assembled after the ribonucleoprotein complex ZBP1 senses IAV infection [62,63].

Host factors could also restrict activation of proinflammatory response and switch to innate immunity. Acetate produced by gut Bifidobacterium pseudolongum NjM1 enhances aggregation of mitochondrial antiviral-signaling protein (MAVS) and production of type I IFN upon GPR43 engagement, which turns down activation of NLRP3 innate immunity in IAV-infected mice [64]. The NLRP3 inhibitor MCC950 and the P2 × 7 receptor inhibitors probenecid and AZ11645373 show beneficial effects in reducing IAV infection-associated hyperinflammation and mortality in mice [65,66]. Due to the intensive vaccine research and the long list of approved direct anti-viral agents for IAV, no clinical trials with NLRP3 inhibitors or drugs that regulate inflammasome activation and downstream signaling have been reported.

### 3.3. Zika Virus

Zika virus (ZIKV) is a member of the Flaviviridae family with positive single-stranded RNA genomes. ZIKV infection during pregnancy is highly linked to abnormalities in congenital brain development and the death of neural progenitors in unborn children [67]. ZIKV-induced systemic inflammation is one important factor that triggers the death of neural progenitors. There is a strong correlation between activation of the NLRP3 inflammasome and serum levels of growth-differentiation factor 3, which regulates early development of embryos in ZIKV-infected pregnant women [68]. ZIKV infection also induces death of macrophages depending on assembly and activation of the NLRP3 inflammasome and it-induced cleavage of GSDMD [69]. No specific drug or vaccine has been approved for Zika virus. Blocking the transmission of circulating virus particles would reduce the infection rate in an outbreak [70].

### 3.4. Other RNA Viruses

The NLRP3 inflammasome can be stimulated by both signal 1 and signal 2, as well as other signals that lead to activation of these signals [2]. Activation of the NLRP3 inflammasome also directly or indirectly involves proinflammatory response, cell death, and pathogenesis related to infection with other RNA viruses. For example, serum and tissue levels of NLRP3 significantly correlate with liver pathology in patients infected with hepatitis C virus [71,72]. The 2B protein of echovirus 11 facilitates assembly of the NLRP3 inflammasome and the subsequent release of IL-1β in murine macrophages [73]. Porcine reproductive and respiratory syndrome virus forms the replication–transcription complex with the help of host endoplasmic reticulum-resident protein TMEM41B, which then stimulates dispersion of the trans-Golgi network and NLRP3-dependent pyroptotic cell death [74]. Elevated ROS and potassium efflux in Mayaro virus (MAYV) infection induce activation of NLRP3 but not AIM2 inflammasome in mice. Patients infected with MAYV also exhibit significantly increased serum levels of proinflammatory cytokine IL-18 and IL-1β compared with healthy individuals [75]. RNA and NS2B protein of foot-and-mouth disease virus trigger assembly of the NLRP3 inflammasome through increased intracellular ion concentration, which depends on the transmembrane region of the NS2B protein that is essential for formation of the ion channel on the host membrane [76]. Thrombocytopenia syndrome virus primes assembly of NLRP3 inflammasome through elevated levels of oxidized mitochondrial DNA and its release into the cytosol [77].

Measles virus is a negative single-stranded enveloped RNA virus that belongs to the genus Morbillivirus and family Paramyxoviridae. The innate immunity triggered by measles virus infection is closely related to the NLRP3 inflammasome assembly and activation and production of TNF-α and IL-6 independent of RNA replication [78]. In contrast, the NLRP3 inflammasome activation in peste des petits ruminants virus, another negative single-stranded RNA virus, is highly dependent on transcription and translation of its genome [79]. The different determinant factor remains largely unknown. However, common features that trigger assembly and activation of the NLRP3 inflammasome have not been reported among these virus types. The distinct determinant factor(s) remain to be elucidated. Most of these studies were performed in isolated murine primary macrophages; further study on patient samples is needed to clarify whether activation of the NLRP3 inflammasome leads to innate immune or inflammatory response for antiviral therapy.

Some RNA viruses utilize assembly and activation of the NLRP3 inflammasome as a checkpoint to escape from immune surveillance. The capsid protein of hepatitis E virus (HEV) and integral HEV viral particles activate transcription of NLRP3 through NF-κB signaling in primary macrophages to antagonize IFN response-driven innate immunity [80]. However, inhibition of the NLRP3 inflammasome with MCC950 or the caspase 1 inhibitor VX-765 significantly increases HEV RNA levels [81], suggesting the contradictory effect of NLRP3 inhibition in HEV infection and the associated inflammatory response. Human T lymphotropic virus type 1 (HTLV-1) is a leukemogenic retrovirus with a single-stranded positive-sense RNA genome. HTLV-1 p12 protein expression leads to inactivation of the NLRP3 inflammasome and expression of the “don’t-eat-me” signal CD47 on cell surface in primary monocytes [82]. Modulation of NLRP3 inflammasome activation by HTLV-1 infection generates escape from immune surveillance and chronic inflammation. The evaluation of inflammasome inhibitors and other therapies is largely hampered by the lack of appropriate animal models to mimic viral infection and associated immune response.

## 4. NLRP3 Inflammasome in DNA Viral Infection

### 4.1. Herpes Simplex Virus 1

Herpes simplex virus 1 (HSV-1) is a dsDNA virus that is 152-kb long. HSV-1 has a high prevalence in humans and is the primary cause of severe inflammation in several types of diseases such as sporadic viral encephalitis [83]. The dsDNA genome of HSV-1 can directly activate the stimulator of IFN response cGAMP interactor 1 (STING), leading to interaction between STING and NLRP3 and translocation of NLRP3 to the endoplasmic reticulum. The ER-resident NLRP3 is deubiquitinated, promoting assembly of the NLRP3 inflammasome and subsequent cleavage of GSDMD and procaspase 1 and release of IL-1β [84]. HSV-1 infection-induced cell death is highly related to activation of the NLRP3 inflammasome in microglia in encephalitis [85,86]. The oncoherpesvirus Epstein–Barr virus (EBV) coordinates activation of several inflammasomes, including NLRP3, to switch to its replicative phase. The NLRP3 inflammasome activated by EBV then depletes the heterochromatin-inducing epigenetic repressor TRIM28 through activated caspase 1, leading to the transcription of proteins required for the replicative phase of EBV [87]. However, efficient replication of HSV-1 requires inhibition of the AIM2 inflammasome by the viral tegument protein VP22 [88]. It is reasonable that HSV-1 must shut down the AIM2 inflammasome to evade the protective innate immune response. The contradictory activation of the NLRP3 inflammasome required for viral replication may not be related to its primary function in innate immunity but rather to the associated changes in endosomal composition [89,90] required for viral replication and budding.

### 4.2. Other DNA Viruses

Activation of the NLRP3 inflammasome is triggered by the expression of hepatitis B virus (HBV) proteins in liver macrophages [91], African swine fever virus infection in monocytes and macrophages [92], and adenovirus-based vaccines in neutrophils [93]. However, the detailed mechanisms that trigger the assembly of the NLRP3 inflammasome upon infection of these viruses remain unresolved. Probably, the comparative study between a real viral infection and a corresponding vaccine without replication ability could shed light on the different function of inflammasome activation.

## 5. Other Inflammasomes

### 5.1. AIM2 Inflammasome

The AIM2 inflammasome first identified in hematopoietic cells senses dsDNA from microbial or host mitochondrial DNA. The AIM2 inflammasome also contains ASC and procaspase-1, the assembly of which is crucial for activation of innate immunity in human papillomaviruses, pseudorabies virus, vaccinia virus, and mouse cytomegalovirus [94,95]. Assembly and activation of the AIM2 inflammasome is identified in infections by RNA viruses such as SARS-CoV-2 and IAV but not other types of dsDNA viruses [44,96]. Therefore, specific dsDNA motifs, methylation of CpG islands of host DNA, or GSDMD-dependent transcription of viral genomes may trigger activation of the AIM2 in infections by these viruses [95,96]. Activation of the AIM2 inflammasome is always associated with the NLRP3 inflammasome in viral infections [97]; therefore, whether activation of the AIM2 inflammasome is the critical point for hyperinflammation or is the consequence of mitochondrial damage and release of mitochondrial DNA triggered by the NLRP3 inflammasome remains to be investigated.

### 5.2. NLRP1 Inflammasome

The central function of NLRP1, the germline-encoded pattern recognition receptor identified to form an inflammasome in 2002 [98], is only now being revealed in epithelial barrier tissues. Upon infection with Semliki Forest virus (SFV) with a positive-stranded RNA genome, NLRP1 undergoes self-cleavage by its function-to-find domain and leads to the generation of two chains of non-covalently associated polypeptides. In addition, unlike other inflammasomes, NLRP1 triggers procaspase 1 cleavage through its C-terminal CARD domain after direct binding to the double-stranded RNA (dsRNA) intermediate formed during SFV replication [99]. NLRP1 activation is also cleaved by 3CL proteases from several coronaviruses, including SARS-CoV-2 in infected pulmonary epithelial cells. The cleaved C-terminal CARD domain of NLRP1 assembles the inflammasome with caspase 1 and induces the release of mature IL-6 through the cell membrane form formed by the SARS-CoV-2 protease NSP5 cleaved gasdermin E [100]. The dsDNA virus Kaposi’s sarcoma-associated herpesvirus Orf63 protein as a viral homolog of human NLRP1 blocks assembly of the NLRP1 inflammasome [101]. Taken together, these results suggest that the NLRP1 inflammasome triggers innate immunity to clear viral infection and inhibit viral persistence.

### 5.3. NLRP6 Inflammasome

The cytosolic innate immune sensor NLRP6 recruits ASC and procaspase 1 or caspase 11 to assemble the inflammasome upon PAMP or DAMP stimulation, similar to NLRP3. The activated NLRP6 inflammasome then triggers the release of IL-18 and IL-1β, which initiates an inflammatory response or drive pyroptotic cell death of infected cells [102]. The expression of NLRP6 is detected in the liver and lung but is the highest in the intestine. Thus, the NLRP6 inflammasome is primarily responsive to changes in the gastrointestinal inflammatory milieu.

Intraperitoneal infection with encephalomyocarditis virus and murine norovirus 1 leads to increased mortality and viremia in NLRP6-/- mice. NLRP6 engages with the viral RNA helicase DHX15, which then recruits MAVS to trigger the type I/III interferon response in the intestine [103]. Ligand stimulation, such as infection with rotavirus or murine hepatitis virus A59, promotes liquid–liquid phase separation (LLPS) of NLRP6. The polylysine sequence (Lys350-354) of NLRP6 is critical for LLPS of NLRP6, which recruits ASC and promotes inflammasome assembly. The cell wall component of Gram-positive bacteria and NLRP6 ligand lipoteichoic acid could also trigger LLPS of NLRP6, condensation of NLRP6 with DHX15 on dsRNA, and the subsequent IFN response [104]. The liquid-like condensed phase could explain the versatility of NLRP6 to different stimulators. The LLPS may also serve as a common feature for the activation of other inflammasomes, such as the NLRP3 inflammasome, which disassemble the trans-Golgi network [89,90,105].

### 5.4. NLRP9b Inflammasome

Similar to NLRP6, NLRP9b expression is high in ileal epithelial cells. NLRP9b senses rotavirus dsRNA infection by interacting with the RNA helicase DHX9. The NLRP9b-dsRNA-DHX9 complex then recruits ASC and procaspase 1 to initiate inflammasome assembly, which subsequently leads to GSDMD-mediated pyroptotic cell death and release of IL-18 and IL-1β. Activated NLRP9b inflammasome clears rotavirus-infected intestinal epithelial cells and reduces susceptibility of mice to rotavirus replication [106]. NLRP9b-/- mice attenuate the inflammatory response to cecal ligation and puncture-induced acute lung injury [107], suggesting that NLRP9b may function more than a dsRNA virus sensor.

### 5.5. NLRC4 Inflammasome

The essential role of the NLRC4 inflammasome is to maintain normal innate immunity during infection. The knockout of NLRC4 leads to the increased expression of Fas Ligand in dendritic cells in IAV-infected mice, promoting death of IAV-infected CD4+ T cells and decreasing overall survival [108]. The HIV gp41 protein directly induces production of IL-18 through NLRC4, independent of the priming signal that drives NLRP3 inflammasome activation [109]. NLRC4 is also involved in the clearance of rotavirus infection in mice mediated through IL-18 [110]. The expression of IL-18 triggered by NLRC4 directly induces the death of rotavirus-infected intestinal epithelial cells mediated by the cognate non-hematopoietic cell receptor but not GSDMD, interrupting the viral lifecycle and leading to the clearance of rotavirus [111].

Niclosamide is an anthelmintic medication used to control tapeworm infestations and is listed on the World Health Organization’s list of essential medicines. Niclosamide inhibits activation of different inflammasome, including AIM2, NLRP3, and NLRC4 inflammasomes by inducing autophagy. Treatment with niclosamide also significantly reduces replication and activation of inflammasome in SARS-CoV-2-infected primary human monocytes and PBMCs obtained from COVID-19 patients [112]. The pan inhibition of niclosamide to NLRP3 and NLRC4 may be due to the highly similar structure of the NACHT domain of these proteins [113], even though the assembly of the NLRC4 inflammasome is different from that of the NLRP3 inflammasomes (Figure 2).

## 6. Checkpoint for Host Defense and Inflammatory Response

The unique mode of cell death during pathogenic infections such as viruses or bacteria is termed PANoptosis. PANoptosis is central to host defense to eliminate infected cells, for example SARS-CoV-2, HSV-1, and IAV [44,62]. Alternatively, cells that undergo massive PANoptosis trigger cytokine release syndrome and destroy host immune immunity [6]. Conventional IFN therapy fails to restrict SARS-CoV-2 infection because IFN-β induced by SARS-CoV-2 contributes to the systemic inflammatory response and cell death dependent on the innate immune sensor ZBP1. The Zα domain of ZBP1 interacts with RIPK2, caspase 6, and caspase 8 and induces PANoptotic cell death [114]. TNF-α and INF-γ are synergistic in the overall inflammatory response during infection with SARS-CoV-2 and lead to PANoptosis of infected cells via RIPK1/FADD/caspase 8 signaling [115]. This suggests that monotherapy to inhibit NLRP3 activity may not provide effective protection against severe viral infection; however, combination therapy may improve overall efficacy in the fight against SARS-CoV-2.

Checkpoints that regulate host defense and hyperinflammation may be conditional. For example, NLRP12 forms a complex with caspase 8, ASC, and RIPK3, termed the PANoptosome, upon stimulation with a combination of heme and various PAMPs. Knockout of NLRP12 reduces hemolytic disease and associated pathology [116]. In contrast, NLRP12 attenuates antiviral immunity in vesicular stomatitis virus infection by mediating ubiquitination and degradation of RIG-I dependent on TRIM25 and RNF125 [117]. These controversial results may be due to the differences in multiplicity of infection (MOI), which is directly related to the severity of disease induced by viral infection. These results highlight the importance of combination therapy with antiviral therapy.

Time of intervention and cell-type specificity are also critical in maintaining the balance between positive and negative regulatory loops. The proinflammatory cytokines such as IL-1β, IL-18, and chemokines secreted by macrophages are required for the migration of neutrophils and dendritic cells to the site of initial infection [118,119]. The blockade of the common downstream factors such as GSDMD and IL-1 has shown efficacy in pilot studies. Dimethyl fumarate, the FDA-approved drug for multiple sclerosis, induces succination of GSDMD, blocking the interaction of GSDMD with caspases and subsequent cell death [120]. Dimethyl fumarate is currently in clinical trials for multiple sclerosis patients infected with SARS-CoV-19 (Table 1) [121,122]. The different function of inflammasomes in myeloid cells and other types of immune cells must be taken into account when developing therapies targeting the inflammasomes. The primary function of AIM2 in myeloid cells is activation in response to stimulation by foreign dsDNA, whereas AIM2 is highly expressed and is required to suppress autoimmunity in regulatory T cells [123].

Available antiviral agents directly target viral proteins, such as neuraminidase, M2 protein, and viral RNA polymerase of IAV, inhibiting various steps in the viral lifecycle instead of controlling the interaction of the virus with the host. Therefore, restriction of viral infection-induced cell death by targeting host factors that promote the switch between inflammatory response and innate immunity, factor(s) controlling conclusion, inflammatory response, and promotion of alternatively activated or M2 macrophages, is also critical to control severe infection-triggered morbidity and mortality. For example, nicotinamide riboside (NAD+) that suppresses neuroinflammation in transgenic Alzheimer’s mice [124] could attenuate neuronal death and cortical thickness in mice with ZIKV infection [125]. Differences between host factors across the species may also provide valuable clues for the development of novel therapies. For example, the human counterpart to bat ASC2 is a natural negative regulator of inflammasome activation (bat ASC2). There are only four critical residues, E10, R37, C61, and G77, that differ between human and bat that could reverse the function of ASC2 to immune suppression in multiple viral infections [126].

## 7. Future Prospective

In this review, we summarized an overview of the current understanding of inflammasomes in sensing viral infection and associated pathology. Targeting the activation of inflammasomes, such as the NLRP3 inflammasome, has proven to be a therapeutic target for the control of viral infection due to its exclusive assembly and activation upon viral infection. However, due to the lack of appropriate animal models and robust characterization of inflammasome assembly during infection with emerging viruses, particularly in the developing countries, the contribution of less-characterized inflammasomes remains largely unexplored. Further studies are needed to clarify whether inflammasome activation is the primary cause of virus-induced pathology or a subsequent alteration of primary antiviral immunity. The development of inhibitors against other inflammasomes has lagged far behind that of NLRP3. The repurposing of clinically used drugs would benefit outbreaks such as COVID-19. In addition, drugs that could overcome emerging viral mutations and the development of pan-inhibitors targeting downstream of the inflammasome would prepare for pandemics. Due to the complexity of the host response to viral infection and its built-in ability to evade immune surveillance, a more comprehensive investigation of the critical function of inflammasomes in viral infection may allow the development of effective treatment strategies for the management of viral infection, especially severe disease such as that induced by infection with pathogenic viruses such as SARS-CoV-2.

## Figures and Tables

**Figure 1 viruses-15-01451-f001:**
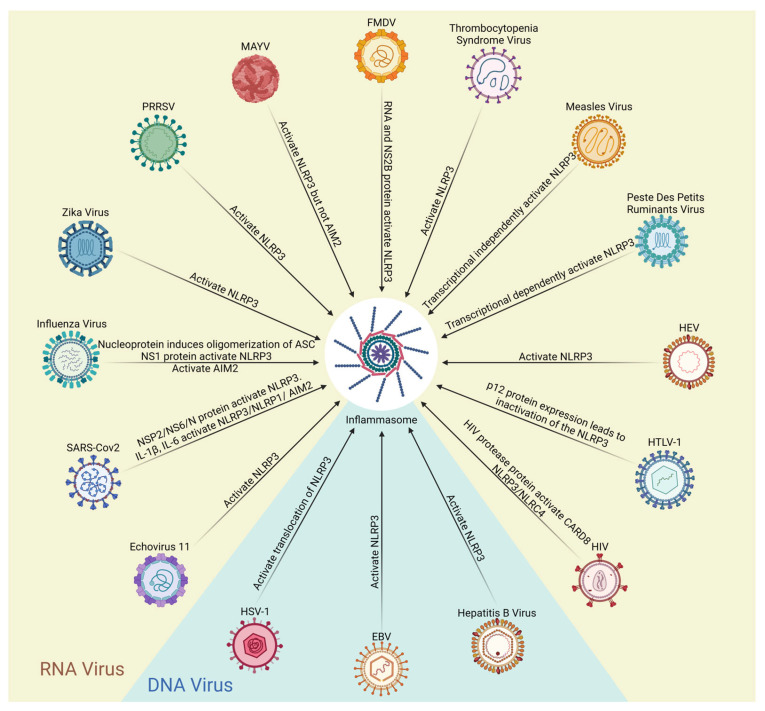
Comparison of different inflammasomes. The PYD domain and the CARD domain of ASC serve as anchor points for the assembly of inflammasomes. However, NLRC4 itself carries the CARD domain, so ASC is not required to form the NLRC4 inflammasome. The figure was created in BioRender.

**Figure 2 viruses-15-01451-f002:**
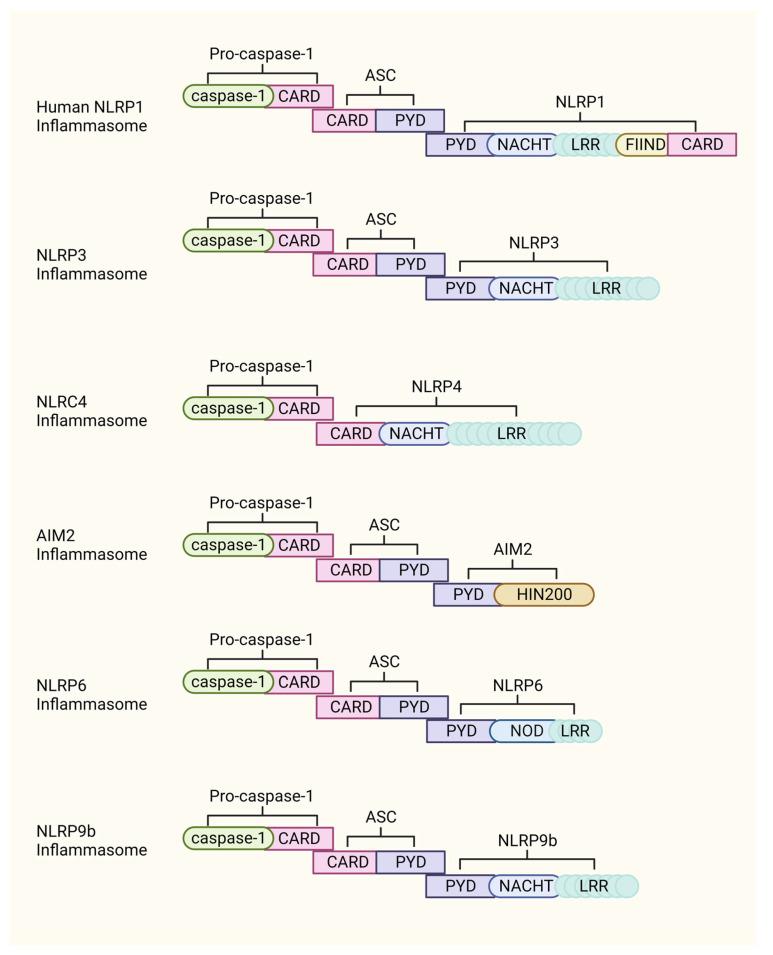
Compositions of different inflammasomes. Most of the identified inflammasomes are composed of NLR proteins, with the exception of NLR4, which recruits ASC by interacting with the PYD domain. The PYD and CARD domains of ASC serve as anchor points for subsequent recruitment of procaspase 1 to complete inflammasome assembly. NLRC4 itself carries the CARD domain, so it interacts directly with procaspase 1 and ASC is not required to form the NLRC4 inflammasome. The figure was created in BioRender.

**Figure 3 viruses-15-01451-f003:**
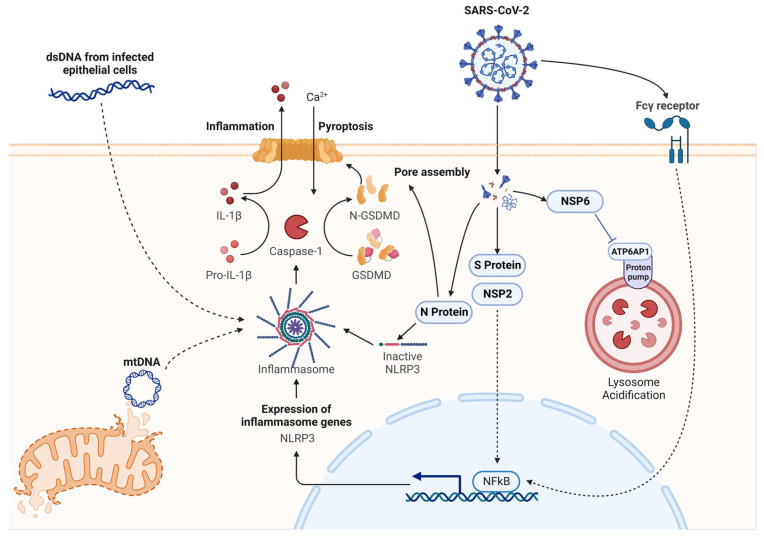
SARS-CoV-2 activates the NLRP3 inflammasome through multiple pathways. SARS-CoV-2 infects macrophages through the Fcγ receptor. Viral proteins, such as S protein and NSP2, also stimulate NLRP3 transcription dependent on NF-κB signaling. N protein of SARS-CoV-2 directly interacts with NLRP3 to induce inflammasome formation and subsequent inflammatory response. dsDNA released from infected epithelial cells and mtDNA released from damaged mitochondria can also activate the inflammasome. The figure was created in BioRender.

**Table 1 viruses-15-01451-t001:** Summary of viruses that activate inflammasome.

Name	Inflammasome	Features of Activation
Positive-sense single-stranded RNA		
HIV	NLRP3/NLRC4/CARD8	HIV protease protein cleaves CARD8. HIV gp41 protein directly induces production of IL-18 through NLRC4.
Zika Virus	NLRP3	There is a correlation between NLRP3 and growth-differentiation factor 3, which regulates early development of embryos in ZIKV-infected pregnant women.
Porcine Reproductive and Respiratory Syndrome Virus	NLRP3	It forms the replication–transcription complex with the help of TMEM41B, which then stimulates NLRP3-dependent pyroptosis.
Mayaro virus	NLRP3	It activates NLRP3 but not AIM2. It also increases the production of IL-18 and IL-1β.
Foot-And-Mouth Disease Virus	NLRP3	RNA and NS2B protein activate NLRP3.
Hepatitis E Virus	NLRP3	Capsid protein and integral viral particles activate transcription of NLRP3 through NF-κB in primary macrophages to antagonize IFN response.
Human T Lymphotropic Virus Type 1	NLRP3	Expression of p12 protein inhibits the NLRP3 inflammasome.
Semliki Forest Virus	NLRP1	NLRP1 undergoes self-cleavage by its function-to-find domain and leads to the generation of two chains of non-covalently associated polypeptides.
Encephalomyocarditis Virus	NLRP6	Intraperitoneal infection with it leads to increased mortality and viremia in NLRP6-/- mice.
Murine Norovirus 1	NLRP6	Intraperitoneal infection with it leads to increased mortality and viremia in NLRP6-/- mice.
Murine Hepatitis Virus A59	NLRP6	Infection with it promotes liquid–liquid phase separation of NLRP6.
SARS-CoV-2	NLRP3/NLRP1/AIM2	NSP2/NS6/N protein activates NLRP3. IL-1β and IL-6 activate NLRP3/NLRP1/AIM2.
Negative-sense single-stranded RNA		
Thrombocytopenia Syndrome Virus	NLRP3	It primes assembly of NLRP3 inflammasome through elevated levels of oxidized mitochondrial DNA and its release into the cytosol.
Measles Virus	NLRP3	It transcriptionally independently activates NLRP3.
Peste Des Petits Ruminants Virus	NLRP3	It transcriptionally dependently activates NLRP3.
Influenza Virus	NLRP3/AIM2	Nucleoprotein induces oligomerization of ASC. NS1 protein activates NLRP3.
Double-stranded RNA		
Rotavirus	NLRC4/NLRP6/NLRP9b	NLRC4 induces death of infected intestinal epithelial cells, leading to clearance of rotavirus. Infection with it promotes liquid–liquid phase separation of NLRP6. NLRP9b senses rotavirus dsRNA infection by interacting with the RNA helicase DHX9 and clears rotavirus-infected intestinal epithelial cells, reducing susceptibility of mice to rotavirus replication.
Herpes Simplex Virus 1	NLRP3	It activates translocation of NLRP3
EBV	NLRP3	The NLRP3 inflammasome activated by EBV depletes the heterochromatin-inducing epigenetic repressor TRIM28, leading to transcription of proteins required for the replicative phase of EBV
African Swine Fever Virus	NLRP3	It activates NLRP3 in monocytes and macrophages and adenovirus-based vaccines in neutrophils
Circular DNA not fully double-stranded		
Hepatitis B Virus	NLRP3	Hepatitis B virus proteins activate NLRP3 in liver macrophage.

**Table 2 viruses-15-01451-t002:** Clinical trials with agents targeting inflammasome for viral infection.

Drug	Clinical Trial Identifier and Phase of Study	Dosage	Disease	Expected Outcome or Results
Targeting NLRP3				
Metformin Glycinate	NCT04625985 (phase 2)	620 mg bid for 14 days	COVID-19 and Severe Acute Respiratory Syndrome Secondary to SARS-CoV-2	Viral load, incidence of adverse events, and changes in laboratory results
Dapansutrile (OLT1177)	NCT04540120 (phase 2)	4 × 250 mg dapansutrile capsules b.i.d. for 14 days with an initial (first) dose of 8 × 250 mg (2000 mg) administered at the study site on Day 1 (Day 1 dose may be 3000 mg)	Moderate COVID-19 symptoms and early cytokine release syndrome (CRS) in patients with confirmed SARS-CoV-2 infection and moderate symptoms	COVID-19-related hospitalization after enrollment or both (1) worsening or persistence of shortness of breath and (2) oxygen saturation less than 92% on room air at sea level or need for supplemental oxygen to achieve oxygen saturation of 92% or greater.
DFV890	NCT04382053 (phase 2)	50 mg was administered orally or nasogastrically twice per day (b.i.d)	COVID-19 pneumonia	Disease severity, serum C-reactive protein levels, clinical status over time, requirement for mechanical ventilation for survival.
Targeting IL-1				
Canakinumab	NCT04362813 (phase 3)	450 mg for body weight of 40 < 60 kg, 600 mg for 60–80 kg or 750 mg for >80 kg	Cytokine release syndrome in patients with COVID-19-induced pneumonia	Number of responders who survived without requiring invasive mechanical ventilation from Day 3 to Day 29.
Canakinumab	NCT04510493(phase 3)	Body weight adjusted dose in 250 mL 5% dextrose solution i.v. over 2 h	Canakinumab in patients with COVID-19 and type 2 diabetes	Survival time, ventilation-free time, ICU-free time, and hospitalization time.
Targeting GSDMD				
Disulfiram	NCT04594343 (phase 2)	500 mg orally once daily for 14 days	Patients with moderate COVID-19	Time to clinical improvement.
Dimethyl fumarase	NCT04381936(phases 2 and 3)	UK adults ≥18 years old only (excluding those on ECMO). 120 mg every 12 h for 4 doses followed by 240 mg every 12 h by mouth for 8 days (10 days in total).	Reduce the risk of dying for patients hospitalized with COVID-19 receiving oxygen.	All-cause mortality, duration of hospitalization, and endpoint of death or need for mechanical ventilation or extracorporeal membrane oxygenation.

## Data Availability

No new data were created or analyzed in this study. Data sharing is not applicable to this article.

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
