# Peer review of "Targeting Inflammasome Activation in Viral Infection: A Therapeutic Solution?"

_viruses, 2023, doi:10.3390/v15071451_

Round 1
Reviewer 1 Report
This manuscript mainly describes the inflammasome activation during several viral infection. Although potentially important and meaningful, there are several points need to be improved greatly, as outlined below.
1、More relevant figures and tables are required, for example, there is a lack of corresponding charts to support the discussion of DNA virus infection and inflammasome, and the description of the charts is too simple.
2、The manuscript likes a simple list of published literature, without systematic and comprehensive comments, and it also lacks the author's evaluation and discussion.
3、The future development and difficulties of this field are not described and discussed, and the authors do not analyze the development with the unique perspective.
4、It is recommend to add more discussion of viral infections and excessive inflammatory responses under inflammasome activation.
no
Author Response
Reviewer #1
- More relevant figures and tables are required, for example, there is a lack of corresponding charts to support the discussion of DNA virus infection and inflammasome, and the description of the charts is too simple.
Response: More relevant figures and tables would help to better summarize the content of this article. We've added Figure 1 and Table 1 to summarize viral infection and the inflammasome, and Table 2 for the NLRP3 inhibitors in clinical trials for SARS-CoV-2. We have also updated the description of Figures 2 and 3 (originally Figures 1 and 2).
- The manuscript likes a simple list of published literature, without systematic and comprehensive comments, and it also lacks the author's evaluation and discussion.
Response: We agree with this reviewer and have highlighted the evaluation and discussion in blue throughout the revised manuscript, especially the intense discussion of the therapeutic potential of NLRP3 inhibitors (section 2.5, page 1-11) and host innate immunity and inflammatory response (section 6, page 16).
- The future development and difficulties of this field are not described and discussed, and the authors do not analyze the development with the unique perspective.
Response: We believe this issue is of critical importance to the future development of the field. The future development and difficulties in the development of inflammasome inhibitors are discussed in Section 7 (pages 16-17).
- It is recommend to add more discussion of viral infections and excessive inflammatory responses under inflammasome activation.
Response: We agree with this reviewer that viral infection and excessive inflammatory responses are critical to the therapeutic potential of inflammasome inhibitors. Further discussion of viral infection and excessive inflammatory response under inflammasome activation has been added in Section 6 (page 16).
Reviewer 2 Report
Deng et al. provide a comprehensive review on inflammasome activation limited by few RNA viruses and DNA virus. It is well written and organized. The role of NLRP3 is well characterized and has potential of therapeutic interventions. However, I feel, this review could have been better by paying much attention on less characterized inflammasome and raising some outstanding questions to the field which is yet missing.
Author Response
Reviewer #2
- This review could have been better by paying much attention on less characterized inflammasome and raising some outstanding questions to the field which is yet missing.
Response: We agree with this reviewer that this is indeed an open question. Due to the limitations of available model systems for less characterized inflammasomes, we have discussed this question. For example, lines 174-180 on page 9, lines 481-489 on pages 16-17, and lines 524-527 on page 16.